# Optical coherence tomography (OCT) in unconscious and systemically unwell patients using a mobile OCT device: a pilot study

Xiaoxuan Liu [1,2,3] , Aditya Uday Kale [2] , Nicholas Capewell,[1] Nicholas Talbot,[1] Sumiya Ahmed,[1] Pearse A Keane,[3,4] Susan Mollan,[1,5] Antonio Belli,[1,6,7] Richard J Blanch,[1,8] Tonny Veenith,[1,7] Alastair K Denniston[1,2,3,4,5]

XL and AUK are joint first authors.

For numbered affiliations see end of article.

**Correspondence to**
Professor Alastair K Denniston;
a.denniston@bham.ac.uk

## ABSTRACT

**Objective** This study aims to evaluate the feasibility of retinal imaging in critical care using a novel mobile optical coherence tomography (OCT) device. The Heidelberg SPECTRALIS FLEX module (Heidelberg Engineering, Heidelberg, Germany) is an OCT unit with a boom arm, enabling ocular OCT assessment in less mobile patients.

**Design** We undertook an evaluation of the feasibility of using the SPECTRALIS FLEX for undertaking ocular OCT images in unconscious and critically ill patients.

**Setting** This study was conducted in the critical care unit of a large tertiary referral unit in the United Kingdom.

**Participants** 13 systemically unwell patients admitted to the critical care unit were purposively sampled to enable evaluation in patients with a range of clinical states.

**Outcome measures** The primary outcome was the feasibility of acquiring clinically interpretable OCT scans on a consecutive series of patients. The standardised scanning protocol included macula-focused OCT, OCT optic nerve head (ONH), OCT angiography (OCTA) of the macula and ONH OCTA.

**Results** OCT images from 13 patients were attempted. The success rates of each scan type are 84% for OCT macula, 76% for OCT ONH, 56% for OCTA macula and 36% for OCTA ONH. The overall mean success rate of scans per patient was 64% (95% CI 46% to 81%). Clinicians reported clinical value in 100% scans which were successfully obtained, including both ruling in and ruling out relevant ocular complications such as corneal thinning, macular oedema and optic disc swelling. The most common causes of failure to achieve clinically interpretable scans were inadequately sustained OCT alignment in delirious patients and a compromised ocular surface due to corneal exposure.

**Conclusions** This prospective evaluation indicates the feasibility and potential clinical value of the SPECTRALIS FLEX OCT system on the critical care unit. Portable OCT systems have the potential to bring instrument-based ophthalmic assessment to critically ill patients, enabling detection and micron-level monitoring of ocular complications.

## Strengths and limitations of this study

► This is the first study demonstrating the capabilities of the FLEX optical coherence tomography (OCT) device in critical care, conducted in one of the largest critical care centres in Europe.

► This feasibility study uses both qualitative and quantitative data to first report image success, and second to provide information regarding image optimisation techniques for both OCT and OCT angiography.

► In addition to scan success and image optimisation techniques, we have included a case series investigating the strengths and limitations of using this novel OCT device in a range of patients with varied clinical states, providing an insight into the potential value of this device in critical care and ward-based patients.

► A key limitation of our study is the cohort size of 13 participants, and although this limits statistical power this cohort provides a valuable insight into the value of mobile OCT devices.

## INTRODUCTION

Optical coherence tomography (OCT) is a non-invasive, light-based imaging technique which enables in vivo, cross-sectional imaging of structures, including human tissues, at high resolution.[1] Its medical application is primarily in retinal imaging where current OCT devices can now accurately resolve the individual retinal layers and provide automated measures of retinal thickness estimated to the nearest micron. Additionally, eye-tracking overlays enable consecutive images to be digitally subtracted allowing detection of changes of even a few microns. It is thus an extremely powerful tool both for diagnosis and monitoring of diseases related to the retina, the optic nerve and other visualisable structures within the eye. A major advance of the last couple of years has been

OCT angiography (OCTA), in which the detection of the phase shift of erythrocytes travelling through the retinal capillaries enables a detailed map of the retinal capillaries, and for the first time a real-time non-invasive map of retinal ischaemia.[2]

The use of ophthalmic OCT and OCTA to detect both physiological and pathological changes could be highly valuable in a critical care setting. One example of its potential utility is mapping the retinal vessels for non-invasive estimations of the microcirculation. It has been shown that retinal blood flow, as measured by fluorescein angiography, exhibits prolonged retinal arterial filling time in patients with lower cardiac index.[3] The same study found microvascular abnormalities such as retinal microaneurysms and retinal haemorrhages (findings common to diabetes and hypertension) in patients with severe sepsis. While fluorescein angiography has its uses, it carries significant risks such as anaphylaxis. It is also not suitable for patients with renal impairment. OCTA on the other hand is non-invasive and fast. This makes it ideal for monitoring the physiological status of a patient with serial scans. Another example of where OCT is valuable is its ability to measure changes in the retinal nerve fibre layer (RNFL). There are a number of neurological conditions in which RNFL changes have been detected, such as RNFL thinning in chronic traumatic encephalopathy following traumatic brain injury and RNFL thickening seen in idiopathic intracranial hypertension.[4–6] OCT-derived measurements could provide surrogate markers of systemic disease which can guide treatment and give valuable prognostic information. So far, this has not been possible due to limitations in the design of current hardware. Most commercially available OCT platforms are static, table-top-mounted machines which scan the patient in an upright, seated position. The patient places their head on a headrest in front of the OCT lens and is asked to fixate as instructed by the operator. This conventional set-up is most suited for patients who are alert, compliant and comfortable at a normal seated height and with a good degree of neck flexion. Topical mydriatics are often given to dilate the pupils for a wider aperture through which the image can be acquired; however, good-quality OCT scans can be acquired in undilated patients.[7] The only other commercially available alternative to the standard table-mounted OCT system is a handheld OCT (Envisu C2300, Leica Microsystems, Wetzlar, Germany); however, handheld devices are lacking in stability and are more sensitive to motion artefact and alignment issues.[8 9]

The Heidelberg SPECTRALIS FLEX module (Heidelberg Engineering, Heidelberg, Germany) is a newly available modified OCT machine designed for use in a variety of clinical settings. In addition to being trolley-based (similar to a portable X-ray machine), it has a boom arm, designed to enable ocular OCT assessment in less mobile patients. The use of ocular OCT imaging as a diagnostic or monitoring tool for systemic conditions is in its infancy, and there are no studies to our knowledge evaluating its role in a critical care setting. This study evaluates the feasibility of using the Heidelberg SPECTRALIS FLEX module in systemically unwell patients in a ward environment.

## METHODS
### Study design and setting
This was a cross-sectional evaluation of the feasibility of using the SPECTRALIS FLEX for undertaking clinically useful ocular OCT images in unconscious and critically ill patients in a major critical care unit at University Hospitals Birmingham NHS Foundation Trust. The critical care unit is a tertiary referral unit with 68 level-3 equivalent beds and is one of the largest critical care units in Europe.

This study was carried out from January to August 2018.

### Outcome measures
The primary outcome of this evaluation was the feasibility of acquiring clinically interpretable OCT scans on a consecutive series of patients, matched against a standard protocol. Acquired scans were reviewed by two ophthalmology specialists experienced in interpreting retinal OCT images. This was reported as percentage of patients for whom clinically interpretable scans were obtained for each modality and percentage of patients for whom the complete scan protocol was obtained. A 100% success rate was defined as all scans successful in both eyes.

### Scan protocol
The standard scanning protocol included four scans: standard macula-focused OCT, OCT optic nerve head (ONH), OCTA of the macula and OCTA of the ONH. This protocol was attempted for both eyes of each patient. The OCT scans used were the Heidelberg Engineering preset 'Fast Macula' and ONH (RNFL) OCT scans. The Fast Macula scan is composed of 25 B scans at automatic real-time (ART) setting 9 (9 images averaged) over an area of $5.7\,mm^2$. The ONH scan is composed of a single B scan ring around the ONH, with an ART setting of 100. The OCTA scans were composed of 512 B scans, totalling an area of $2.8\,mm^2$, with an ART setting of 5.

### Statistical analysis
Descriptive statistics are provided with feasibility and clinical value being reported as percentages; tests of statistical significance are not relevant to this study.

### OCT assessment with the SPECTRALIS FLEX
The optical system attached to the Heidelberg SPECTRALIS FLEX module is the same as the table-top platform, equipped with the 'OCT2 system' to provide both OCT and OCTA. The major difference to the table-top system is that the acquisition lens is mounted on a flexible 'boom' arm which can be extended away from the main body of the module to approximately 100 cm (figure 1). The flexible arm can be configured and tightened at different pivotal points along its length, thereby accommodating patients in a variety of different positions during scanning. The main body of the FLEX is mounted

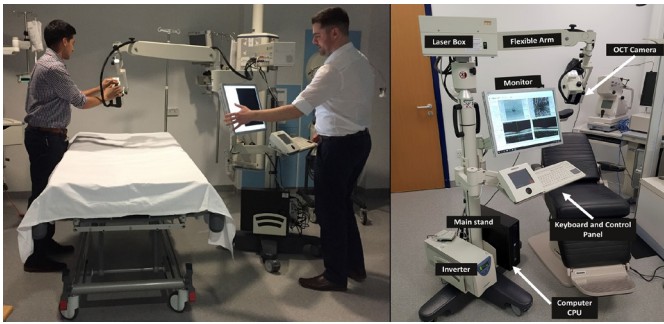

**Figure 1** The Heidelberg SPECTRALIS FLEX module positioning as demonstrated by authors (AUK and NC). CPU, central processing unit; OCT, optical coherence tomography.

on wheels enabling portability of the module. The FLEX system is usually mains-powered but can operate for approximately 20 min on battery reserve when needed.

We found several adjustments relating to position of the machine and preparation of the patient which significantly improve ease of image acquisition as described in the following two sections. Operators were already proficient in operating table-top OCT scanners; however, training over approximately 2 hours was conducted to allow the staff to familiarise themselves with operation and manoeuvring of the FLEX module.

### Manoeuvrability and positioning

The device can be positioned comfortably at the bedside, and its extendable arm gives sufficient reach over a hospital bed (including critical care beds). Our approach to positioning the device and its operators around the bed space is shown in figure 1. Operator 1 is positioned over the patient, holding the OCT camera piece in front of the patient to make small adjustments to alignment during acquisition if required. Operator 2 operates the desktop interface to lock and acquire the image and change settings in between scans using the keyboard. In an unconscious patient, having an additional operator standing behind the patient's head to hold it in position and to lift the eyelid is advantageous.

### Preparation of the patient

Pupillary dilation was undertaken unless clinically contraindicated. We found that pupillary dilation had a much greater impact on quality of scans than with conventional OCT. The larger aperture partly compensates for problems with maintaining optical alignment which may arise from several issues including lack of forehead and chin support, systemically unwell patients losing fixation due to fatigue, or unconscious/delirious patients not fixating at all. Ample verbal prompting and reassurance were given during longer acquisition modalities (notably OCTA) to help the patient maintain focus. Ocular surface lubricants were given between scan acquisitions to improve the ocular surface, a common issue in unconscious patients which can impact the quality of scans.

### Patient and public involvement

As this was a pilot study evaluating novel technology, patients and the public were not involved in the design or conduct of this study.

## RESULTS

### Technical feasibility

Feasibility was assessed against the completion of the full protocol of structural and vascular assessment of the retina and optic nerve (comprising OCT macula, OCT ONH, OCTA macula and OCTA ONH). OCT images from 13 patients were attempted, with at least one clinically interpretable OCT being achieved in 92% (12 of 13 patients). None of the included patients had known existing ocular pathology. The success rate of each scan type from the scan protocol are 84% successful for OCT macula, 76% for OCT ONH, 56% for OCTA macula and 36% for OCTA ONH. Completion of the full protocol was possible in 31% (4 of 13 patients). The mean scan protocol success rate per patient was 64% (95% CI 46% to 81%). Table 1 summarises the conditions under which scans were obtained.

### Clinical value

Clinical value was reported in 100% cases where interpretable scans were achieved. In these scans the clinical value was reported as 'ruling out' serious complications of the underlying systemic disease; in the remainder the clinical value was either diagnostic or the ability to quantitatively monitor retinal or ONH status.

### Case examples

The following cases highlight the clinical value in four different contexts and demonstrate the potential clinical value of the OCT modalities available.

#### Case 1: postoperative oesophagectomy

Patient 1, who was admitted for two-stage oesophagectomy for an oesophageal adenocarcinoma, was scanned at day 1 postoperatively, while making a good recovery. At the time he was able to sit up in bed, was alert, compliant and able to maintain good fixation, enabling successful acquisition of the full scan protocol (figure 2). Serious complications including retinal ischaemia and retinal vascular occlusive disease were ruled out based on the scans.

#### Case 2: sepsis

Patient 2 was admitted due to accidental paracetamol overdose with secondary renal failure, pulmonary failure and sepsis. At the time of scanning the patient was sedated and intubated. The patient was scanned in a semirecumbent position in bed, with eyelids held open by one of the scan operators (figure 3). Additional technical challenges were (1) the poor quality of the ocular surface due to poor lid closure and inadequate lubrication resulting in corneal epitheliopathy (exposure keratopathy), and (2)

**Table 1** Summary of scan acquisition in all cases, success rates for completing the whole protocol and problems encountered

| Case number | Diagnosis | Patient's age | Patient's sex | Position of patient | Conscious/unconscious | Scans acquired (n) | Whole protocol achieved? | Problems encountered |
|---|---|---|---|---|---|---|---|---|
| 1 | Post-operative oesophagectomy | 55 | Male | Sat up | Alert | 8 | Yes | No problems of note. |
| 2 | Sepsis | 64 | Female | Supine in bed | Unconscious and intubated | 3 | No | Poor ocular surface, drifting gaze, difficult positioning due to airway. |
| 3 | Traumatic brain injury | 33 | Female | Semi-recumbent in bed | Unconscious and intubated | 4 | No | Unable to dilate both eyes (to avoid interfering with neurological observations). |
| 4 | Toxic epidermal necrolysis (TEN) | 39 | Female | Semi-recumbent in bed | Semi-sedated and delirious | 0 | No | Ocular surface involvement of TEN with corneal erosions and delirious patient. |
| 5 | Post-operative oesophagectomy | 75 | Male | Semi-recumbent in bed | Alert | 8 | Yes | No problems of note. |
| 6 | Post-operative oesophagectomy | 84 | Female | Sat up in chair | Alert | 3 | No | Patient was scanned upright in a chair with minimal head support. |
| 7 | Post-operative oesophagectomy | 64 | Female | Semi-recumbent in bed | Alert | 8 | Yes | No problems of note. |
| 8 | Traumatic brain injury | 36 | Male | Semi-recumbent in bed | Unconscious and intubated | 3 | No | Patient needed eyelids held open, difficulty achieving optimum alignment. |
| 9 | Post-operative oesophagectomy | 66 | Male | Semi-recumbent in bed | Alert | 4 | No* | Scans were difficult to obtain even with dilation due to poor fixation and the patient's lack of sleep. |
| 10 | Post-operative oesophagectomy | 56 | Male | Sat up in chair | Alert | 6 | No* | All scans achieved without dilation. |
| 11 | Neutropaenic sepsis | 53 | Male | Supine in bed | Unconscious | 8 | Yes | Patient scanned with lids held open. 3 operators were required. |
| 12 | Post-operative oesophagectomy | 56 | Male | Semi-recumbent in bed | Alert | 3 | No | Patient was amblyopic in the right eye with poor fixation. |
| 13 | Post-operative oesophagectomy | 58 | Male | Sat up in chair | Alert | 5 | No | Patient was in pain during scans with poor fixation. |

*Postoesophagectomy scans were carried out at 24 hours postoperatively in the critical care unit. An asterisk marks where all scans were obtainable at 7 days postoperatively on a normal ward. All patients but one (case 4) had at least one successful scan.

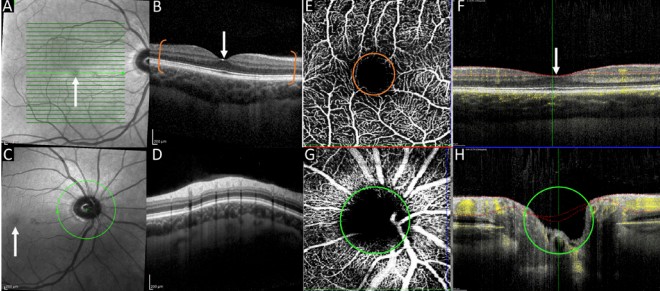

**Figure 2** Case 1. The above composite image includes OCT and OCTA scans showing the retinal layers and retinal vasculature. OCT macula infrared image (A) and cross-sectional view (B). Image B is a cross-section of the macula indicated by the white arrows in images A–C and F. The bracketed area of image B indicates the retinal layers from the inner limiting membrane to the retinal pigment epithelium, below which is the choroid. Scans A and B are most commonly obtained in clinical practice and can be used to exclude retinal disease such as age-related macular degeneration. OCT optic nerve head infrared image (C) and cross-sectional scan (D). OCTA macula en face view (E) and cross-sectional view (F). The circular area in image E marked with an orange border is the foveal avascular zone (a round, capillary free zone) corresponding to the area of the macula. Hyper-reflective areas indicate red blood cell movement (ie, blood flow), with hyporeflective areas indicating a lack of red blood cell movement. Cross-sectional images F and H show blood flow indicated by yellow areas of the OCTA scan optic nerve head en face view (G) and cross-sectional view (H). The optic nerve head in image C corresponds to the hyporeflective area in image G. The optic nerve is highlighted by the green border in images C, G and H. OCT, optical coherence tomography; OCTA, OCT angiography.

patient movement resulting in an alignment artefact in the reconstructed OCTA rendition of the capillary plexi.

### Case 3: traumatic brain injury

Patient 3 was admitted following polytrauma and traumatic brain injury. Scans were acquired while the patient was intubated and sedated. The left eye (scanned eye) was fixed and dilated secondary to traumatic brain injury (figure 4). Optic nerve swelling, retinal detachment and retinal haemorrhage were successfully ruled out in this eye using the OCT scan. The right eye was not dilated using topical mydriatics so as not to interfere with the patient's neurological observations; scan quality through the undilated pupil was poor, with acquisition of OCTA not being possible; therefore, imaging in this eye was abandoned.

### Case 4: toxic epidermal necrolysis

Patient 4 was admitted with toxic epidermal necrolysis (TEN) with 60% cutaneous involvement and corneal erosions. This patient was scanned in a supine position. The patient was being weaned off sedative medication and was agitated and delirious, with a Glasgow Coma Scale score of 10 out of 15. A macula-focused OCT was acquired; however, further imaging was abandoned due to quality of scans. An additional scan—anterior segment

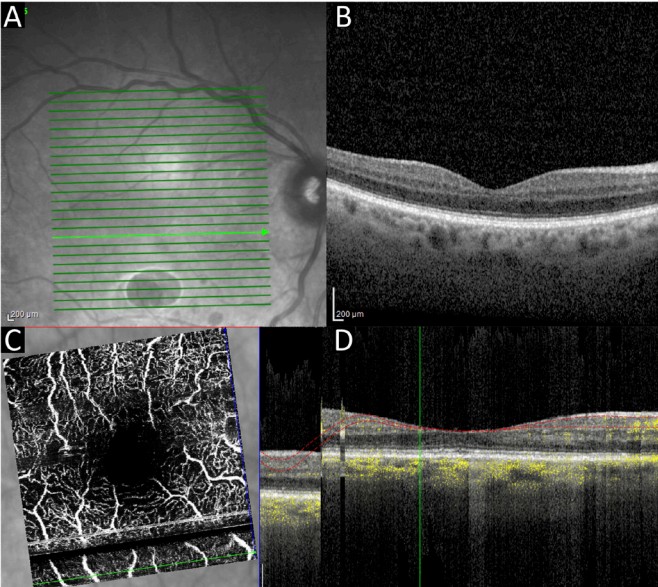

**Figure 3** Case 2. Right eye OCT macula infrared (A) and cross-sectional view (B). OCTA macula en face view (C) and cross-sectional view (D). Poor ocular surface can be seen in the infrared image (A), and an alignment artefact arising from patient movement can be seen as a dark stripe on the OCTAs (C, D). OCT, optical coherence tomography; OCTA, OCT angiography.

(AS) OCT—was performed so as to evaluate the cornea since the primary clinical concern in cases of TEN is of corneal and ocular surface disease. A good-quality, single-section AS-OCT was obtained, and serious corneal thinning was successfully ruled out. The major technical challenges here were the poor ocular surface related to her disease and the low level of patient compliance and fixation (figure 5).

## DISCUSSION

In this prospective evaluation, we demonstrated the feasibility of using the SPECTRALIS FLEX module for undertaking clinically interpretable ocular OCT images

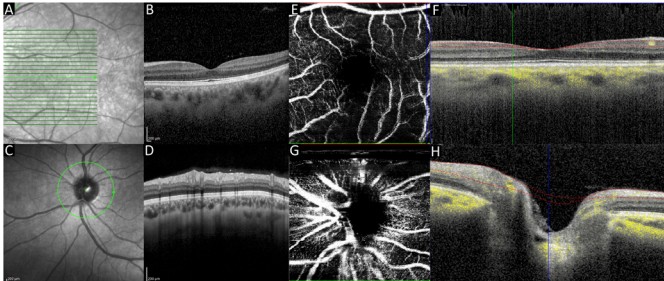

**Figure 4** Case 3. OCT macula infrared image (A) and cross-sectional scan (B). OCT optic nerve head infrared image (C) and cross-sectional scan (D). OCTA macula en face view (E) and cross-sectional view (F). OCTA optic nerve head en face view (G) and cross-sectional view (H). OCT, optical coherence tomography; OCTA, OCT angiography. Green lines in images 4A and 4C indicate areas where corresponding cross-sectional images were obtained.

in unconscious and critically ill patients in a critical care unit. The primary outcome of this evaluation was the feasibility of acquiring clinically interpretable OCT. Our experience is that OCT macula was possible in 84% and OCT ONH in 76% of our cohort. OCTA was more challenging, with success rates of 56% for macula and 36% for ONH.

OCT is already recognised as a vital part of the clinical assessment for most retinal and optic nerve diseases, providing a permanent objective record and quantitative assessment of physiological and pathological status. Its clinical value is not therefore in doubt, but it is not yet clear whether this can be translated into the critical care environment.

The SPECTRALIS FLEX is a novel device. The only report of its use is in an intra-operative setting for acquiring aqueous angiographic images during phacoemulsification.[10] Handheld OCT devices are available, and their use has been described in a range of patients including in children,[11 12] in sedated patients[13] and in the intra-operative environment.[8] The only commercially available handheld OCT device is the Envisu C2300 (Leica Microsystems). This compact device weighs only 1.6 kg and is connected to a desktop system on a mobile cart via a 2 m cable. It can acquire OCT images but does not have OCTA capabilities. In terms of stability and the avoidance of movement artefact, the SPECTRALIS FLEX provides better stability, since operator movement is minimal. However, this does not compensate for patient movement (ie, in the delirious patient) as seen in case 4, resulting in the typical misalignment.

The purpose of this evaluation was to provide a 'real-world' assessment of the feasibility of undertaking OCT using the FLEX system. The completion of the whole protocol was only possible in 31% of patients, which reflects the challenges that need to be overcome in the critical care environment, compared with an outpatient setting. Several reasons for scan failure are common to both environments. They include inability to maintain alignment (commonly acute confusion in critical care; dementia/learning difficulties; nystagmus and poor concentration in clinic); ocular surface disorders (commonly exposure keratopathy in critical care; any corneal pathology in clinic); media opacity (rare cause of failure in critical care; common cause in clinic due to, for example, cataracts); and extreme refractive errors. There are challenges associated with the SPECTRALIS FLEX module even among those who are experienced in using the table-top OCT systems. Our extensive experience has enabled us to refine our technical approach to improve capture, and guidance to help other users is provided as part of our protocol in the Methods section of this manuscript.

Key limitations of this study include first our small sample size of 13 patients, which limits the generalisability of the results; however, even with this small number, we are able to provide proof-of-concept as to the value of the FLEX module in a range of clinical conditions. Second,

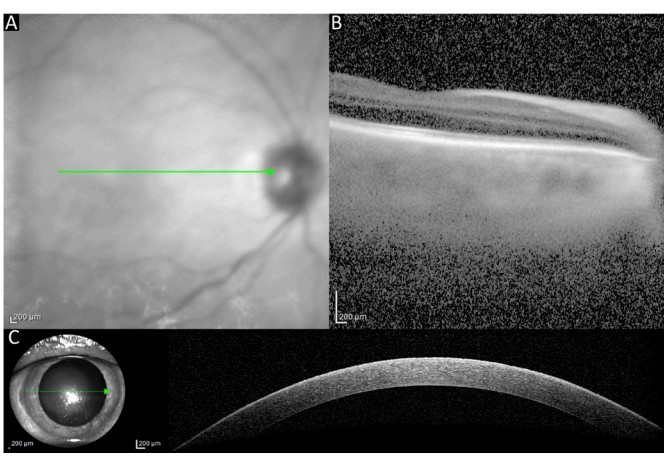

**Figure 5** Case 4. OCT macula (A, B) could only be acquired using a single line scan and the quality was poor. A single line macula was obtained, but the rest of the protocol was abandoned. A single line OCT scan of the anterior segment was obtained to assess integrity of the cornea (C). Green arrows shown in these images represent the area show in the corresponding cross-sectional views.

as a new technique, there may have been a significant 'learning effect' as OCT operators had to learn to overcome the challenges of moving from a stable table-top device in clinic to using the mobile FLEX unit on systemically unwell patients on a critical care unit. It is worth noting however that there was no general trend in the success rate of scans over time. Lastly, scan success was partly dependent on the patient's clinical state, and patient fatigue might reduce the success of scans attempted towards the end of our protocol. This may explain the lower success rate of OCTA ONH scans (which are relatively more demanding and time-consuming); standard OCT scans are unlikely to be affected as these are fast and less demanding for patients.

In our evaluation we found that clinicians reported clinical value in all cases where scans were possible, including both 'ruling-in' and 'ruling-out' relevant ocular complications. These may be sight-threatening complications such as corneal thinning in TEN or severe exposure keratopathy or macular oedema in diabetic eye disease. Alternatively, they may reflect life-threatening central nervous system (CNS) or systemic pathology, such as retinal microvascular changes or optic disc swelling in papilloedema. Although not tested with the SPECTRALIS FLEX module, standard table-top OCT and OCTA have been used in a range of relevant conditions, including raised intracranial pressure, neurodegenerative diseases and dementia.[14–17] Additionally, the retinal vasculature exhibits autoregulation to systemic insults such as hypoxia and has demonstrated quantifiable reductions in vessel density during hypoxic provocation.[18 19] Further work should aim to test the reliability and clinical utility of the FLEX module and compare this with both conventional table-top devices and handheld OCT systems. Further analysis of these devices should be undertaken in the critical care setting

with a longitudinal follow-up to visualise retinal changes in patients with a range of clinical conditions.

## CONCLUSIONS

We have evaluated the feasibility of using a novel, mobile OCT system to obtain retinal scans in unconscious and critically ill patients. Portable OCT systems have the potential to bring objective instrument-based ophthalmic assessment to the bedside, enabling detection and micron-level monitoring of ocular complications which may impact on vision or potentially serve as indicators of systemic health. This prospective evaluation indicates OCT and OCTA are feasible on the critical care unit, providing a promising foundation for the development of a new tool for improved clinical assessment and potential pathophysiological discovery in these patients.

#### Author affiliations
[1]Ophthalmology Department, University Hospitals Birmingham NHS Foundation Trust, Birmingham, UK
[2]Academic Unit of Ophthalmology, Institute of Inflammation and Ageing, College of Medical and Dental Sciences, University of Birmingham, Birmingham, UK
[3]Health Data Research UK, London, United Kingdom
[4]NIHR Biomedical Research Centre for Ophthalmology, Moorfields Eye Hospital NHS Foundation Trust, London, UK
[5]Centre for Rare Diseases, Institute of Translational Medicine, Birmingham, UK
[6]Institute of Inflammation and Ageing, University of Birmingham, College of Medical and Dental Sciences, Birmingham, UK
[7]Department of Critical Care Medicine, University Hospitals Birmingham NHS Foundation Trust, Birmingham, UK
[8]Academic Department of Military Surgery and Trauma, Royal Centre for Defence Medicine, Birmingham, UK

**Acknowledgements** We are grateful to Ali Tafreshi, Christopher Mody and Krysten Williams and the team at Heidelberg Engineering for their expertise and for providing the Heidelberg SPECTRALIS FLEX module for this study.

**Contributors** AKD, PAK, TV, AB, SM and RJB conceived the study. XL, AUK, NC, SA and NT undertook data collection. XL, AUK, RJB and AKD undertook data analysis. AKD, TV and RJB supervised this study. All authors participated in drafting and review of the manuscript.

**Funding** XL, PAK and AKD receive a proportion of their funding from the Wellcome Trust, through a Health Improvement Challenge grant (200141/Z/15/Z). AKD and PAK receive a proportion of their funding from the Department of Health's NIHR Biomedical Research Centre at Moorfields Eye Hospital and UCL Institute of Ophthalmology. TV, NT and AB receive a proportion of their funding from Queen Elizabeth Hospital Birmingham Charities.

**Competing interests** None declared.

**Patient consent for publication** Not required.

**Ethics approval** Approval was granted by University Hospitals Birmingham NHS Foundation Trust for this prospective evaluation as a service evaluation, the purpose being to establish feasibility of introducing the standard of care for ophthalmic assessment using OCT to the local critical care unit.

**Provenance and peer review** Not commissioned; externally peer reviewed.

**Data availability statement** Data are available upon reasonable request.

**ORCID iDs**
Xiaoxuan Liu http://orcid.org/0000-0002-1286-0038
Aditya Uday Kale http://orcid.org/0000-0002-2186-1446

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
