## [Reviewer comments · BMJ Open]

ARTICLE DETAILS

TITLE (PROVISIONAL)	Optical Coherence Tomography (OCT) in Unconscious and Systemically Unwell Patients Using a Mobile OCT Device: A Pilot Study
AUTHORS	Liu, Xiaoxuan; Kale, Aditya Uday; Capewell, Nicholas; Talbot, Nicholas; Ahmed, Sumiya; Keane, Pearse A; Mollan, Susan; belli, antonio; Blanch, Richard J; Veenith, Tonny; Denniston, Alastair K

VERSION 1 – REVIEW

REVIEWER	Bamini Gopinath University of Sydney, Australia
REVIEW RETURNED	23-May-2019

GENERAL COMMENTS	This study aimed to evaluate the feasibility of retinal imaging in critical care using a novel mobile optical coherence tomography (OCT) device. It recruited 13 patients and reported image success, and provided information regarding image optimisation techniques for both optical coherence tomography (OCT) and OCTA. The real novelty of this study lies with the use of an OCT unit with a boom arm, enabling ocular OCT assessment in less mobile patients. 1. It is stated throughout the paper that this is a prospective study and it involved prospective evaluation of OCT images in critically ill patients. If this is the case, what was the follow-up period over which these images were obtained and what during what year(s) was this study conducted. This information needs to be included in the methods and abstract. From what I read it appears to be only a cross-sectional study but this needs to be clarified fully by the researchers.2. How was consent obtained for participation in this study?3. Reading Table 1, it appears the whole protocol was completed only on 3/13 (23%) and not 31% (4/13 patients) as the Authors report. Please clarify.4. A table summarising the clinical characteristics of the patients that were included in the study e.g. age, sex, presence of ocular conditions e.g. macular degeneration, cataracts; would have been useful.5. There are some major drawbacks due to the sample size and sampling method used in this study, for example, there is an inability to generalize research findings and low level of reliability and high levels of bias. These limitations need to be highlighted in the Discussion section of the paper.6. There are several sentences throughout the paper that needs to be moderated. For instance, in the Discussion (lines 405-406): '...providing a powerful new tool for improved clinical assessment and potential pathophysiological discovery in these patients.' Based on the current study alone it is not possible to conclude this, and the authors should reword this statement.
--

REVIEWER	Hanna G. Zimmermann Charité - Universitätsmedizin Berlin, Germany
REVIEW RETURNED	13-Jun-2019

GENERAL COMMENTS	A feasibility study of the application of a movable OCT device in a critical care unit. Though this is an interesting and important new field for OCT and the authors claim it is just ment to be a feasibility study, I am not sure about it's relevance for fellow researchers and clinicians because the results are based on a small and heterogenous cohort which doesn't really allow quantitative analysis. At least some ambiguities should be clarified and some quantitative information should be included.  - The authors should describe the scan protocols in more details: Scan size, number of B-Scans, number of A-scans per B-scan, ART rate. This is important information because even in the normal OCT setup these factors influence the examination time. - Was the order of scans always the same? Isn't it likely then to have a higher success rate in the earlier scans than in the last one? So was the order the reason that OCTA failed ore often, rather than the nature of the scan? - It is unclear whether both or just one eye were scanned, and whether „Whole protocol achieved?“ refers to one eye or both. - The authors claim that “Completion of the full protocol was possible in 31% (4/13 patients). However, regarding to the table this was the case in only 3 patients. Please clarify. - Table 1 would benefit from additional information on whether at least one scan was successfully recorded, or none. - From table 1 we learn that the whole protocol was only achieved in alert patients, it would be interesting to know whether in unconscious and delirious patients there were at least some usable scans. (exact numbers) - The authors mention in the discussion that there is a learning curve with the method. Was the lack of experience perhaps the reason for failure of some examinations in the beginning of the studies? Could the authors try to quantitatively analyze the association of the operator's experience and success rates - The study provides very little evidence that the Spectralis Flex is superior regarding image quality in comparison to hand-held OCT in these patients, can the authors provide any evidence for that?
---

VERSION 1 – AUTHOR RESPONSE

Reviewer 1 response:

We appreciate the time taken to review our paper, and we thank reviewer 1 for recognising the value of our study. Please see our responses to the reviewer comments below:

1. It is stated throughout the paper that this is a prospective study and it involved prospective evaluation of OCT images in critically ill patients. If this is the case, what was the follow-up period over which these images were obtained and what during what year(s) was this study conducted. This information needs to be included in the methods and abstract. From what I

read it appears to be only a cross-sectional study, but this needs to be clarified fully by the researchers.

- a. **Thank you for your comment regarding our study design, we agree that our design is cross-sectional. We have amended the working in our methods section and abstract.**
2. How was consent obtained for participation in this study?
 - a. **This study was approved by the University Hospitals Birmingham NHS Foundation Trust clinical governance department as a service evaluation for assessing the clinical utility of a routinely used ophthalmic imaging device in a critical care setting. Local institutional review recognised that as the OCT scan is a routinely used, non-invasive imaging method to enhance standard care, consent was deemed to not be required.**
3. Reading Table 1, it appears the whole protocol was completed only on 3/13 (23%) and not 31% (4/13 patients) as the Authors report. Please clarify.
 - a. **We appreciate this being pointed out and have rectified this error. We have ensured that the numbers are consistent with our reported table.**
4. A table summarising the clinical characteristics of the patients that were included in the study e.g. age, sex, presence of ocular conditions e.g. macular degeneration, cataracts; would have been useful.
 - a. **We thank reviewer 1 for this valuable suggestion. We have included age and sex characteristics in table one. We have added a sentence to clarify that none of the patients had known ocular conditions (page 12, line 262).**
5. There are some major drawbacks due to the sample size and sampling method used in this study, for example, there is an inability to generalize research findings and low level of reliability and high levels of bias. These limitations need to be highlighted in the Discussion section of the paper.
 - a. **We agree and have included a paragraph in the discussion to highlight the limitations of our study (page 19 lines 402-414). We aim to demonstrate the scope and limitations of the FLEX device in this exploratory study.**
6. There are several sentences throughout the paper that needs to be moderated. For instance, in the Discussion (lines 405-406): '...providing a powerful new tool for improved clinical assessment and potential pathophysiological discovery in these patients.' Based on the current study alone it is not possible to conclude this, and the authors should reword this statement.
 - a. **We agree and have moderated how we describe the value of OCT in this context, to acknowledge that this is still exploratory. We hope that our research provides a strong foundation for further exploratory research in the field of ocular imaging in critical care.**

Reviewer 2 response:

A feasibility study of the application of a movable OCT device in a critical care unit. Though this is an interesting and important new field for OCT and the authors claim it is just meant to be a feasibility study, I am not sure about its relevance for fellow researchers and clinicians because the results are based on a small and heterogeneous cohort which doesn't really allow quantitative analysis. At least some ambiguities should be clarified and some quantitative information should be included.

We thank reviewer 2 for their valuable suggestions regarding our manuscript. We appreciate that our sample size is limited, however this is a pilot study to demonstrate the scope of using a mobile OCT device in a critical care environment. In particular, we wanted to explore the full range of patients and clinical statuses one might find in critical care, hence we believe that a heterogeneous cohort may be advantageous in this instance. We have addressed the following comments, and aimed to clarify details that reviewer 2 has kindly highlighted.

1. The authors should describe the scan protocols in more details: Scan size, number of B-Scans, number of A-scans per B-scan, ART rate. This is important information because even in the normal OCT setup these factors influence the examination time.
 - a. **We thank reviewer 2 for this suggestion and have included these details in the methodology in lines 201-207 (page 9).**
2. Was the order of scans always the same? Isn't it likely then to have a higher success rate in the earlier scans than in the last one? So was the order the reason that OCTA failed ore often, rather than the nature of the scan?
 - a. **We recognise this, and have now acknowledged it more clearly in our discussion (page 19, line 409-414). We often perform the OCTA of the optic nerve head last, as this can require repositioning the patient's head. For pragmatic reasons we chose to attempt OCTA macula and the two OCT scans before continuing with the OCTA optic nerve head scan.**
3. It is unclear whether both or just one eye were scanned, and whether „Whole protocol achieved?“ refers to one eye or both.
 - a. **We thank reviewer 2 for pointing this out, and have included the details in the scan protocol section (page 9, line 201). We have also defined what we mean by 100% success rate (page 8, line 195).**
4. The authors claim that “Completion of the full protocol was possible in 31% (4/13 patients). However, regarding to the table this was the case in only 3 patients. Please clarify.
 - a. **We appreciate this being pointed out and have rectified this error to be consistent with table 1.**
5. Table 1 would benefit from additional information on whether at least one scan was successfully recorded, or none.
6. From table 1 we learn that the whole protocol was only achieved in alert patients, it would be interesting to know whether in unconscious and delirious patients there were at least some usable scans. (exact numbers)
 - a. **In response to comments 5 and 6, we thank the reviewer for their valuable suggestions with regards to table 1. We have updated table one with the exact numbers of scans acquired, and have added information to the legend regarding whether or not at least one scan was achievable for patients.**
7. The authors mention in the discussion that there is a learning curve with the method. Was the lack of experience perhaps the reason for failure of some examinations in the beginning of the studies? Could the authors try to quantitatively analyze the association of the operator's experience and success rates.
 - a. **We thank the reviewer for this suggestion. We did not perform a formal usability analysis. However, we have now included information regarding our operator training (page 10, line 229).**
8. The study provides very little evidence that the Spectralis Flex is superior regarding image quality in comparison to hand-held OCT in these patients, can the authors provide any evidence for that?
 - a. **We thank the reviewer for pointing this out. Since this study did not specifically include a handheld OCT comparator, we have removed this statement.**

VERSION 2 – REVIEW

REVIEWER	Hanna Zimmermann Charité - Universitätsmedizin Berlin
REVIEW RETURNED	02-Oct-2019
GENERAL COMMENTS	My comments have been addressed.